# Molecular Subtype Prediction for Breast Cancer Using H&E Specialized Backbone

**Anonymous Author**                          AN8@SAMPLE.COM

**Anonymous Author**                          AN9@SAMPLE.COM

## Abstract

Identifying the molecular markers to categorize breast cancer is one of the key steps in determining the prognosis and treatment strategy. The standard clinical practice is to do this analysis based on multiple immunohistochemistry (IHC) stainings for each biomarker, which is expensive and inconsistent when lacking resources. In this work, we investigated the predictiveness of morphological characteristics of Hematoxylin and Eosin (H&E) stained tissues for molecular subtype analysis, as an initial step for direct treatment response prediction based on H&E whole slide images (WSI). Transfer learning using backbones pre-trained on natural images is a common practice to deal with the challenge of lack of large and precisely annotated datasets. However, using pre-training on natural images is not optimal for clinical images. To deal with this challenge and leverage large pools of unlabeled data, we propose to use a specialized backbone pre-trained on H&E WSI in a self-supervised setting, i.e. without any labels. Our experiments show that this backbone is capable of learning discriminating morphological characteristics from H&E images which are well predictive of the molecular subtypes in weakly supervised settings. Also, the network performs better in terms of generalization to unseen data from new scanner types, despite the relatively small size of the dataset used for pre-training the backbone.

**Keywords:** molecular subtypes; H&E; WSI; representation learning

## 1. Introduction

Molecular analysis for breast cancer based on immunohistochemistry staining (IHC) provides important diagnostic, prognostic, and predictive information for clinicians to guide therapy (hormone, targeted or immunotherapy along with surgery, chemotherapy, and radiation). This analysis is done by pathologists based on immunohistochemistry (IHC) staining, which selectively identifies antigens in cells of a tissue section e.g. estrogen receptor (ER), progesterone receptor (PR), and human epidermal growth factor receptor 2 (HER2). IHC stained tissues are often costly, and subject to high variability during preparation and analysis. Hematoxylin and Eosin (H&E) staining on the other hand, which highlights the morphological cellular and architectural features on tissue sections, is less expensive and variable to preparation and analysis.

The connection between molecular analysis and morphological phenotypes has been investigated in recent studies. Rawat et al. (2020) concluded that a combination of both stromal and epithelial areas of the tissue will lead to the final molecular subtype prediction. Heng et al. (2017) showed that poorly differentiated epithelial tubules were prognostic in ER-positive (ER+) breast cancer, while they could not find any prognostic feature in ER-negative (ER-) cases. Naik et al. (2020) discovered that the characteristics associated with low-grade tumors and invasive lobular carcinoma appear more in ER+ samples.

Using deep learning techniques for molecular analysis based on histological phenotypes in H&E images is a challenging task. There is only one label for each whole slide image (WSI) and there are no clear histological characteristics that pathologists could annotate for this task. Annotating all potentially useful cellular or architectural features is also not cheap and practical. Transfer learning and Multiple Instance Learning (MIL) techniques are widely used in such scenarios when limited weakly labeled datasets are available Ilse et al. (2018). Often, the embeddings of all the instances are extracted with networks pre-trained on large scale labeled datasets of natural images (e.g. ImageNet Deng et al. (2009)) and then used in combination with new layers which are tuned for specific tasks at hand. Using such pre-trained networks on natural images is not optimal for H&E images with totally different appearances and characteristics. Thus replacing the backbone with a specialized network trained on these images make sure that meaningful morphological features are used in the downstream task. But the challenge in training such backbones is the availability of large-scale labeled data in this domain. Rawat et al. (2020) proposed the tissue fingerprinting method, which learns from large sets of unannotated H&E WSIs by distinguishing one patient from another. They enforced style invariance by style transfer of images from a reference site to images from other sites. However, this pipeline needs matching cohorts with different stainings which is expensive and not always available.

Recent advances in representation learning with self-supervised techniques Grill et al. (2020); Chen and He (2021); Goyal et al. (2021); Caron et al. (2020) have eliminated the need for annotations since they do not need any labels during training. The main idea in these techniques is to use contrastive training to maximize the similarity between the representations of two augmented views of the same image, regardless of their labels. In this work, we employ such techniques to learn the characteristics of H&E images from a large pool of diverse H&E images without any labels or constraints. Then we use this specialized backbone, in an attention-based MIL technique to predict the molecular subtypes of breast cancer from H&E slides, as assessed by IHC for ER, PR, and HER2. Using an attention-based framework helps in explaining the regions of interest detected by the network for the final decision.

## 2. Method

### 2.1. Representation learning

Unsupervised learning of visual features from large unlabeled sets has had large progress in the computer vision domain. In this work, we start by adapting the method proposed by Grill et al. (2020) (BYOL) to learn characteristics of H&E images from a diverse pool of H&E images. BYOL uses two copies of the same network in parallel which should result in similar representations for two augmented views of the same image. The weights of the target encoder are obtained using the slow-moving average of the weights of the online network with no gradient back-propagation, which prevents the model to collapse to uninformative and trivial solutions.

Having diverse yet realistic augmentations is a key step in training BYOL. Thus, we adapted the augmentations used in Grill et al. (2020) to more realistic augmentations for H&E WSIs. Color and scale play key prognostic and diagnostic roles. We have reduced the range of changes for these augmentations to make sure this information is preserved,

while the network becomes invariant to staining and small-scale variation. The applied augmentations include limited random color augmentations, cropping and scaling, horizontal and vertical flipping, rotation, translation, and Gaussian blurring. We used the standard Resnet18 He et al. (2016) backbone in PyTorch Paszke et al. (2019). Depending on the complexity of the problem and the size of the dataset, a similar approach can be used for training networks with a higher capacity. Larger networks led to overfitting in our experiments.

### 2.1.1. Datasets

The dataset used for training BYOL consists of four subsets taken from the publicly available projects of The Cancer Genome Atlas (TCGA) including TCGA-BRCA, TCGA-LUAD, TCGA-THCA, and TCGA-DLBC. Additionally, we used three private cohorts provided by commercial vendors covering different types of tissue, tumor, and diseases (e.g. breast, lung, thyroid, lymph node and tonsil tissues, follicular lymphoma, diffuse large B-cell lymphoma diseases, etc.). Due to the large size of WSIs, the slides are considered as a large grid of non-overlapping tiles. Each tile is extracted and treated as a separate image in the training setup. Additional pre-processing steps have been applied to make sure the tiles are taken from good quality tissue regions only. Thus, the tiles which are outside of the tissue (in white regions), blurred or include pen-markers or other artifacts are excluded. We have prepared $256 \times 256$ tiles at $0.5$ µ$m$ resolution. For each study, an equal number of tiles from the slides of each study (around 150k tiles per study) are selected to make sure the final dataset is not imbalanced and biased towards a specific slide, tissue type or staining. In total 1045378 tiles are used. A portion of tiles (10%) is selected as the validation set and the rest is used at training. The same loss as the training loss is calculated for the validation set to make sure there is no overfitting during training.

### 2.2. Molecular subtyping using an attention-based MIL framework

An attention-based deep MIL framework Ilse et al. (2018) is used for the downstream task of molecular subtyping from H&E images. The embeddings are generated using two backbones: one pre-trained on H&E tiles (Section 2.1), and one pre-trained on ImageNet dataset Deng et al. (2009). Then, the embeddings are used as input for training the attention layers and the classifier head. The attention-based MIL pooling helps in obtaining a bag representation that is further processed by a bag-level classifier to provide the final score for each slide. After training is done, several attention heatmaps for correctly and wrongly classified samples are generated. The heatmaps are reviewed by a pathologist to get an insight into the decision made by the network.

Since the samples may have different antigens or hormonal receptor status, we can consider four main categories for them including: (I) Her2-, ER+; (II) Her2+, ER-; (III) Her2+, ER+; and (IV) Her2-, ER-, PR- (Triple Negative Breast Cancer (TNBC) where all three antigens are absent). For all the categories, except for the last one, PR can be either present or not. Based on these categories, we designed three main experiments to change the complexity of the classification task depending on how different classes are grouped with each other.

**Experiment 1** Two-class classification between (II) vs. (III) to identify the specific morphological properties of ER;

**Experiment 2** Three-class classification between (I) vs. (IV) vs. (III) and (II) combined;

**Experiment 3** Classification between all four categories ((I) vs. (II) vs. (III) vs. (IV)).

### 2.2.1. Datasets

Several cohorts are used for these experiments, which are specific to the molecular subtyping task and are different from the dataset used for pre-training the backbone (see Section 2.1.1). In addition to the publicly available TCGA-BRCA dataset (Leica Aperio scanner), six private cohorts provided by Commercial Vendors (CVD) including the samples with three different molecular subtypes and scanned with two different scanners (Philips Intellisite Ultra Fast Scanner and Ventana DP200) at 0.25 µm scan resolution are used. In addition, two Internal Datasets called ID1 (Hamamatsu NanoZoomer XR scanner) and ID2 (Ventana DP200 scanner) are used. To test the performance of the models in different scenarios, we have designed two test settings, namely:

**Test 1** 80% of slides from each cohort are used in training and 20% for testing.

**Test 2** 80% of slides from the cohorts TCGA, ID1 and ID2 are used in training and 20% for testing, while all slides from the commercial vendors (CVD) are used for testing.

All stratification was done at the patient level, i.e. no slides or tiles from the same patient appear in both the training and test sets. Table 1 presents an overview of the number of slides per cohort (train/test) per category for each of these settings. In Test 2, the scenario is more similar to a real-world example, where usually algorithms are trained on specific cohorts and then are used, or tested, on independent cohorts. These independent cohorts may contain more variations due to scanners or preparation and staining protocols.

| Cohort | class (I) | class (II) | class (III) | class (IV) | Total |
|---|---|---|---|---|---|
| TCGA-BRCA | 493/121 | 32/11 | 123/29 | 121/33 | 769/194 |
| CVD | 150/36 | 50/12 | 82/26 | 160/40 | 442/114 |
| ID1 | 71/19 | 9/2 | 24/4 | 6/1 | 110/26 |
| ID2 | 939/248 | 118/26 | 168/44 | 513/128 | 1738/446 |
| **Test 1** | 1653/424 | 209/51 | 397/103 | 800/202 | 3059/780 |
| **Test 2** | 1503/574 | 159/101 | 315/185 | 640/362 | 2617/1055 |

Table 1: An overview of the number of slides per cohort used in train/test sets

The tiles are extracted after similar pre-processing steps as mentioned in Section 2.1.1 at 0.50 µm magnification level with $256 \times 256$ resolution. The final tiles are then used to train the network for each combination of the experiments and test settings which results in 6 networks in total. In total 6,024,702 tiles are used for experiment 1 and and 27,713,707 tiles for the other two experiments. For each MIL network several hyper-parameters including the

batch size, the number of samples per bag, learning rate, drop out, weight decay factor and the number of training epochs were tuned using grid-search to give the best area under the receiver-operator characteristics curve (AUC) performance on the 20% validation set. The validation sets were chosen randomly from each training set, also split at the patient level. The best performing networks for each experiment-test combination are then evaluated on their corresponding test sets. The results are presented in Section 3.

## 3. Results and Discussion

An overview of the patient-level AUC scores obtained in each experiment is available in Table 2. For each experiment, the results are obtained using two different backbones (see Section 2.2) and two different test sets. For experiments 2 and 3, which are multi-class, the AUC scores are calculated as the average AUC score of each pairwise class combination. Comparing across different experiments, experiment 1 is trained with fewer data since it is focusing on classifying the Her2+ samples only into two categories (ER+/-). Experiments 2 and 3 could obtain slightly higher performances since the samples are split into finer categories and more data is used during the training. In experiment 2 and 3, the biggest confusion happens within the non-TNBC samples where signal from both Her2 and ER are present. By treating this category as a separate class (class (III)) in experiment 3, the performance decreased due to a higher confusion between the samples of this class and the two other categories (classes (II) and (I)).

| Experiment | Backbone | Test 1 | | Test 2 | |
|---|---|---|---|---|---|
| | | Val AUC | Test AUC | Val AUC | Test AUC |
| Experiment 1 | ImageNet | 0.805 | 0.722 | 0.785 | 0.713 |
| | H&E | 0.813 | 0.715 | 0.818 | 0.674 |
| Experiment 2 | ImageNet | 0.878 | 0.840 | 0.857 | 0.709 |
| | H&E | 0.840 | 0.839 | 0.835 | 0.685 |
| Experiment 3 | ImageNet | 0.836 | 0.799 | 0.821 | 0.678 |
| | H&E | 0.820 | 0.820 | 0.800 | 0.650 |

Table 2: Validation and test AUCs obtained for each experiment and test setting

For all the experiments, the results obtained using our specialized backbone on H&E images are comparable to the results achieved with the backbone pre-trained on ImageNet, even though the size of our dataset is smaller and the training is done without any labels. This shows the great potential of the self-supervised learning techniques in using large pools of unlabeled cohorts for pre-training various backbones to be used in other downstream tasks. In terms of generalization to new unseen data (Test 2), the performances for both backbones are lower than the performances achieved for Test 1. This can be resolved to some extent by using stain normalization, augmentation or domain adaptation techniques.

Furthermore, we show the AUC obtained separately for each cohort in Table 3. Since the ID1 cohort contains too few test slides, the comparison for different settings on this

cohort is not very informative. For the TCGA-BRCA and ID2 datasets we observe that the H&E-based backbone can obtain results on-par with or better than the ImageNet backbone. One interesting observation is the clear advantage of using the H&E backbone on the CVD-Philips datasets in Exp. 2 & Test 2 and Exp. 3 & Test 2. In Test 2, no slides from this type of scanner is included in the training set. So, we could deduce that the H&E backbone is able to generalize better to an unseen scanner type. The ImageNet backbone performs better in the same experimental setting for the CVD-DP200 dataset. However, since ID2 dataset is also obtained using the same scanner, there are many (2617) training examples from the DP200 scanner. Thus, it would be fair to say that it is an easier task.

| Setting | Backbone | TCGA-BRCA | | CVD - DP200 | | CVD - Philips | | ID1 | | ID2 | |
|---------|----------|-----|-------|-----|-------|-----|-------|-----|-------|-----|-------|
| | | Num | AUC | Num | AUC | Num | AUC | Num | AUC | Num | AUC |
| Exp.1 & Test 1 | ImageNet | 40 | 0.639 | 19 | 0.885 | 19 | 0.782 | 6 | 1.00 | 70 | 0.736 |
| | H&E | | 0.649 | | 0.731 | | 0.769 | | 0.25 | | 0.783 |
| Exp.1 & Test 2 | ImageNet | 40 | 0.680 | 85 | 0.703 | 84 | 0.712 | 6 | 0.750 | 70 | 0.758 |
| | H&E | | 0.803 | | 0.653 | | 0.649 | | 0.375 | | 0.782 |
| Exp.2 & Test 1 | ImageNet | 193 | 0.758 | 56 | 0.918 | 56 | 0.929 | 26 | 0.877 | 446 | 0.83 |
| | H&E | | 0.760 | | 0.951 | | 0.923 | | 0.750 | | 0.823 |
| Exp.2 & Test 2 | ImageNet | 193 | 0.755 | 277 | 0.654 | 272 | 0.529 | 26 | 0.920 | 446 | 0.828 |
| | H&E | | 0.735 | | 0.559 | | 0.69 | | 0.879 | | 0.833 |
| Exp.3 & Test 1 | ImageNet | 193 | 0.722 | 56 | 0.898 | 56 | 0.865 | 26 | 0.675 | 446 | 0.802 |
| | H&E | | 0.739 | | 0.897 | | 0.869 | | 0.787 | | 0.783 |
| Exp.3 & Test 2 | ImageNet | 193 | 0.721 | 277 | 0.601 | 272 | 0.538 | 26 | 0.759 | 446 | 0.811 |
| | H&E | | 0.715 | | 0.553 | | 0.639 | | 0.735 | | 0.801 |

Table 3: Test AUCs obtained per cohort and experiment. Num: Number of test slides.

Additionally, we created several attention heatmaps for each experiment by overlaying a mask of attention values per tile on the original image. A sample heatmap is shown in Figure 1. The color spectrum from red to blue corresponds to high to low attention regions. The second row shows a smaller region of the WSI at a higher magnification level, and the second column represents the corresponding heatmaps for the first column. The initial reviews by a pathologist indicate that the network is focusing on biologically meaningful regions such as tumor epithelial cells and collagen-rich stroma for making the final decision. Some categories are easier to identify such as TNBC in which the tumor growth is not affected by Her2 or ER status. On the other hand, based on the literature Freudenberg et al. (2009) Her2 over-expression is a dominant transformation mechanism in tumors, thus the presence of both ER and Her2 makes the discriminatory morphological patterns for HR+ harder to identify. That was also reflected in our experiments. Additionally, based on the attention maps in experiment 1, the pathologist observed that the lymphocytic infiltration in some tumor nests and necrotic areas are informative for differentiating the Her2+, ER- (class (II)) samples from Her2+, ER+ (class (III)) ones.

## 4. Conclusion

Nowadays, large cohorts of weakly labeled or unlabeled digital pathology images are available in clinics. Leveraging this large amount of data using novel techniques such as self-supervised learning can lead to substantial performance improvements in practice. In this

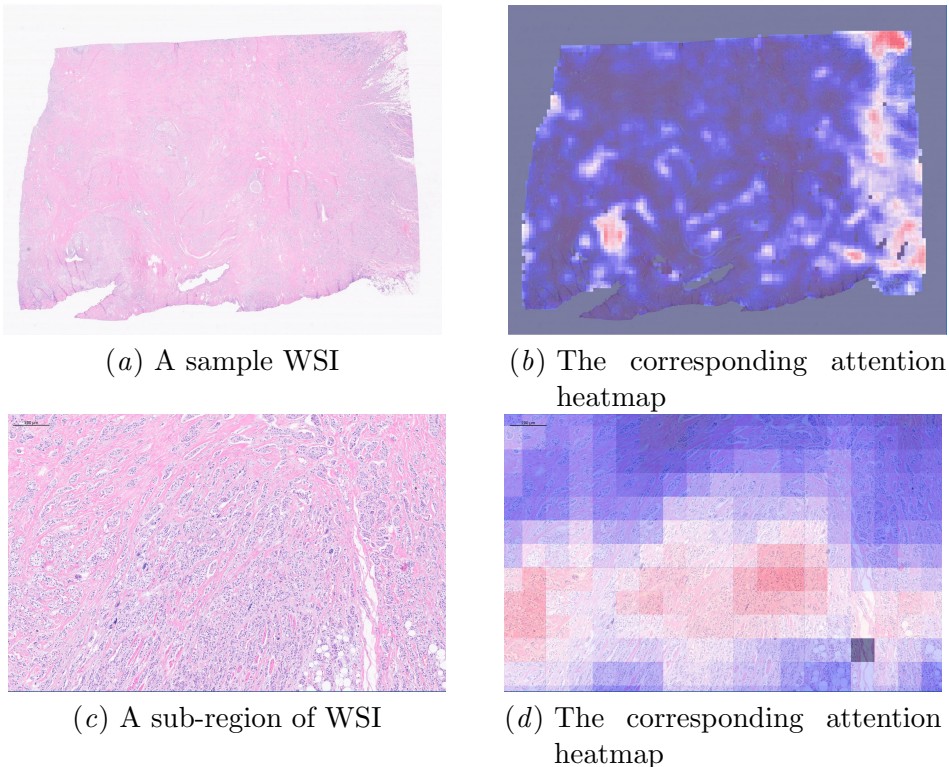

(a) A sample WSI

(b) The corresponding attention heatmap

(c) A sub-region of WSI

(d) The corresponding attention heatmap

Figure 1: A sample attention heatmap obtained in experiment 1 (best viewed in color).

paper, we illustrated a potential application of a self-supervised approach for the downstream task of molecular subtype classification in breast cancer. Our preliminary results show that the specialized backbone pre-trained on H&E images without using any labels is capable of learning biologically meaningful morphological patterns. Since this training regime does not need any labels, further improvements can easily be obtained not only by using a larger and more diverse pool of data but also by using data at multiple scales, which is feasible due to the large size of WSIs. The right balance between the capacity of the network and the size of the dataset needs to be investigated as well. Moreover, the generalization can be improved not only by using stain normalization and more extensive augmentations but also by using data with more diverse stainings scanned with different scanners and with different formats.

Our experiments show a strong indication of the feasibility of molecular subtyping using only H&E images. This is very promising and can provide faster and less expensive information to clinicians compared to using IHC staining which is more expensive, time-consuming, and subject to more variation in preparation and analysis. In the future, we plan to validate the regions of interest selected by the network using the corresponding IHC stained slides when available.

## Acknowledgments

The results published here are in part based upon data generated by the TCGA Research Network: https://www.cancer.gov/tcga.

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
