# OpenReview forum: "Molecular Subtype Prediction for Breast Cancer Using H&E Specialized Backbone"
_MICCAI.org/2021/Workshop/COMPAY — COMPAY 2021_

### Official Review · Reviewer_aK1D · 2021-07-31
**A combination of self-supervised with MIL H&E analyses to predict clinical routine IHC biomarker status**

**Rating:** 7
**Confidence:** 4

**Review:**

The paper is well written, providing good references to the background, and the design of the experiments is comprehensive. The achieved AUC results are promising but the relative low performance in Test 2 calls for additional interpretation. The main issues with the work are:

1. The proposed explanation for the drop in Test 2 test performance is weak "...resolved to some extent by using stain normalization, augmentation or domain adaptation techniques", in particular since most of those augmentation techniques are already used in the experiments.
2. The pathologist interpretation of the results remain unclear, in particular the functional/causal relation of the observed morphologies with the predicted ER/HER2 status. The spatial scale of the attention maps does not allow a detailed interpretation since no individual cells are resolved by the attention mechanism used.
3. The clinical utility is questionable in the light of highly optimized routine ER/PR/HER2 IHC diagnostics.The benefit for the quality of the diagnosis in terms of predicting response or survival has not been discussed.

---

### Official Review · Reviewer_yGQF · 2021-08-17
**Molecular Subtype Prediction for Breast Cancer Using H&E Specialized Backbone**

**Rating:** 5
**Confidence:** 5

**Review:**

The authors in this paper utilize an of-the-shelf MIL approach to identify molecular biomarkers (ER, PR, HER2) from H&E WSIs.

The paper does not have any technical novelty but trying to address an important clinical problem.

There is no motivation/reason in the paper for considering these three experiments!
It seems that the is no validation (tunning) set is considered for doing the experiments. It brings the validity of the results in question.
There is no confidence interval for experiments. It would be good to report the results of multiple runs for each experiment.
It would be good if we have AUC for each class, How does the model perform for under-represented classes (II and III).

---

### Decision · Program_Chairs · 2021-08-25

Accept